# Coffee Consumption and the Progression of NAFLD: A Systematic Review

**DOI:** 10.3390/nu13072381

**Published:** 2021-07-12

**Authors:** Rebecca Sewter, Susan Heaney, Amanda Patterson

**Affiliations:** 1School of Health Sciences, College of Health, Medicine and Wellbeing, University of Newcastle, Callaghan, NSW 2308, Australia; rebecca.sewter@uon.edu.au (R.S.); susan.heaney@newcastle.edu.au (S.H.); 2Department of Rural Health, College of Health, Medicine and Wellbeing, University of Newcastle, Port Macquarie, NSW 2444, Australia; 3Priority Research Centre for Physical Activity and Nutrition, University of Newcastle, Callaghan, NSW 2308, Australia

**Keywords:** systematic review, NAFLD, coffee, fibrosis, cirrhosis

## Abstract

Non-alcoholic fatty liver disease (NAFLD) is the leading cause of chronic liver disease in developed countries. Coffee is one of the most consumed beverages in the world and has been shown to be beneficial in limiting progression in chronic liver disease in general. However, research surrounding the impact of coffee consumption on NAFLD progression is limited. This systematic review aimed to investigate the relationship between coffee consumption and the progression of liver disease, specifically for cases of NAFLD. MEDLINE, EMBASE, CINAHL, the Cochrane Library, and Scopus were searched for published studies that evaluated the effects of coffee consumption on the progression of NAFLD. The results are presented in a narrative synthesis with principal summary measures, including odds ratios, *p*-values, and differences in mean coffee intake in relation to severity of NAFLD. Five studies met the inclusion criteria and were included in this review. There was no trial evidence among NAFLD patients, rather all studies were of a cross-sectional design. Using the Academy of Nutrition and Dietetics Quality Criteria Checklist, four studies received a positive rating, with the remaining study receiving a neutral rating. Overall, four out of the five studies reported a statistically significant relationship between coffee consumption and the severity of fibrosis. Methods around capturing and defining coffee consumption were heterogeneous and therefore an effective dose could not be elucidated. Results suggest that higher coffee consumption is inversely associated with the severity of hepatic fibrosis in individuals with NAFLD. However, further research is required to elucidate the optimum quantity and form/preparation of coffee required to exert this hepatoprotective role.

## 1. Introduction

Non-alcoholic fatty liver disease (NAFLD) is the leading cause of chronic liver disease in developed countries and is estimated to affect approximately 30% of adults in Australia [1]. NAFLD refers to the accumulation of excess lipids in the liver (>5% of total liver weight) in individuals with no significant alcohol consumption (<30 g daily for men and <20 g daily for women) and no other known secondary causes [2,3]. NAFLD develops more commonly in men and at an older age, with the disease diagnosed predominantly between age 40–60 years [3,4]. It is strongly associated with type 2 diabetes mellitus, hypertension, insulin resistance, and hyperlipidemia [2,4,5]. As such, it has been shown to be associated with lifestyle factors such as obesity, poor diet, and lack of exercise [5]. An emerging area of interest has been the relationship between coffee intake and NAFLD, however, while there is evidence to suggest that coffee intake may reduce the risk of developing NAFLD, little is known about the role of coffee in the progression of NAFLD.

NAFLD includes a range of disease states progressing from simple steatosis to non-alcoholic steatohepatitis through to fibrosis and cirrhosis [6]. Early recognition and treatment of NAFLD is crucial to prevent the progression from the early stages to fibrosis and/or cirrhosis as these conditions cause damage to the liver and affect its ability to function normally [7]. Current treatment focuses on trying to reduce hepatic steatosis as well as address and treat the metabolic risk factors for NAFLD, such as hypertension and dyslipidemia [8]. While numerous pharmacological agents are available to treat these metabolic risk factors, evidence supporting the use of pharmacological interventions for NAFLD itself are lacking [7,9]. Therefore, there are currently no pharmacological treatments approved to treat NAFLD [7,9]. Instead, clinical evidence supports the role of lifestyle intervention as a primary therapy for the management of NAFLD [7,9,10].

As there is a limited consensus on the best dietary treatment for NAFLD, research exploring the use of new therapeutic agents for NAFLD treatment is increasing [11]. One area of focus has been the effect of coffee on NAFLD progression [12]. Numerous studies conducted over the past 30 years have demonstrated the beneficial health effects of coffee on chronic liver disease [12,13]. There is Level 1 evidence to support the role of coffee consumption in reducing the risk of hepatic fibrosis and cirrhosis [14,15]. Similarly, there is evidence reporting the beneficial health effects of coffee consumption, ranging from improved liver enzymes, in the form of lower levels of gamma-glutamyl transpeptidase, serum alkaline phosphatase, and alanine aminotransferase, through to improvements in hepatic steatosis and fibrosis, and a reduced risk of cirrhosis and hepatocellular carcinoma [16]. This meta-analysis concluded that there are benefits to encouraging daily intake of coffee for patients with chronic liver disease [16].

In more recent years, studies have been conducted to investigate whether coffee demonstrates similar health effects for individuals with NAFLD. Many epidemiological studies have reported an inverse relationship between coffee consumption and hepatic fibrosis in individuals with NAFLD, indicating that these beneficial effects of coffee on the liver are independent of the etiology [12,17]. A 2010 case-control study on individuals with NAFLD found reduced hepatic steatosis among coffee drinkers versus non-coffee drinkers [18]. A 2012 cross-sectional study found that higher coffee intake resulted in significantly lower risk of advanced fibrosis among individuals with NAFLD [19]. Similarly, a 2012 case-control study found that a high coffee intake was associated with lower grade NAFLD [20].

Coffee is a multifarious combination of biologically active compounds, including kahweol, cafestol, and, in particular, caffeine [21]. Caffeine is a natural stimulant found in coffee, tea, soft drinks, and energy drinks and whose main biological effect is the competitive antagonism of the adenosine receptor [22]. Many studies attribute these beneficial health effects of coffee to the caffeine contained within the beverage [23]. However, decreased hepatic fibrosis appears to be specific to coffee and does not seem to be shared by other caffeinated beverages [24,25]. A 2017 systematic review found that a higher intake of decaffeinated coffee was associated with a decreased risk of hepatocellular carcinoma [26]. Studies investigating the effects of decaffeinated coffee on liver health, in particular NAFLD, are limited. However, a 2019 animal study found that consumption of decaffeinated coffee protects the liver of mice from the development of early stages of NAFLD [27]. The results of these studies indicate that there may also be a role played by one or more of the other substances present in coffee, however, more human studies are needed [23,24,27].

Multiple individual studies have supported the association between coffee consumption and reduced hepatic fibrosis among individuals with NAFLD and conclude that regular consumption of coffee should be encouraged, however, the exact influence of coffee on NAFLD progression remains unclear [17,25]. In addition, the amount and form of coffee that must be consumed to achieve this hepatoprotective effect has not been determined [13,17,25]. Current systematic reviews on this topic are limited and focus primarily on the relationship between coffee consumption and the risk of development of NAFLD. The aim of this systematic review was to investigate the relationship between coffee consumption and the progression of NAFLD to fibrosis and/or cirrhosis.

## 2. Materials and Methods

The Preferred Reporting Items for Systematic Reviews and Meta-Analyses (PRISMA) guidelines were used to conduct this systematic review [28]. The protocol for this review was registered on the PROSPERO international prospective register of systematic reviews: https://www.crd.york.ac.uk/prospero/display_record.php?RecordID=192814 (accessed on 15 July 2020).

### 2.1. Eligibility Criteria

Eligibility criteria for this review were developed using the PICOS (Population Intervention Comparison Outcome Study Design) model. Included studies were those assessing the progression of NAFLD to hepatic fibrosis and/or cirrhosis in relation to coffee consumption. This review considered studies that met the following inclusion criteria: 1. females and males of any age with NAFLD; 2. intervention studies such as cross-over trials, randomised controlled trials (RCT), quasi-RCT, and pre- and post-studies, as well as observational studies such as cohort studies, case control studies, and cross-sectional studies; 3. studies that utilised hepatic imaging or liver biopsies to evaluate NAFLD progression; 4. any type and quantity of coffee consumption were eligible for inclusion, however, studies that did not differentiate between coffee intake and overall caffeine intake were excluded. Systematic reviews and/or meta-analyses were excluded, as were animal studies.

### 2.2. Information Sources and Search Strategy

An electronic literature search was conducted for articles published in English on five separate databases (MEDLINE, EMBASE, CINAHL, the Cochrane Library, and Scopus) with the help of an academic librarian using key search terms. The chosen search terms were coffee or caffein* AND ‘NAFLD’ or ‘non-alcoholic fatty liver disease’ or ‘liver fibrosis’ or ‘hepatic fibrosis’ or ‘liver cirrhosis’ or ‘hepatic cirrhosis’ or ‘fatty liver’ or ‘NASH’ or ‘non-alcoholic steatohepatitis’. References were managed using the online systematic review manager Covidence [29] and Endnote X9 referencing software [30].

### 2.3. Study Selection

Duplicates were removed and abstracts and titles of all articles were screened by two reviewers (RS and SH), with any conflicts resolved by a third reviewer (AP). Studies recognised as meeting the eligibility criteria, or those that were undetermined, had the full article retrieved. All full texts were independently reviewed by two reviewers (RS and AP) against eligibility criteria, and any conflicts were resolved by a third reviewer (SH). The reason for a study’s exclusion from the review was noted. All full text articles that met inclusion criteria moved to the quality assessment and data collection stages.

### 2.4. Quality Assessment

The Academy of Nutrition and Dietetics Quality Criteria Checklist was used to evaluate the methodological quality of the included studies [31]. This method assigns studies a rating of positive, neutral, or negative based on 14 main questions designed to assess the strength of the research design, validity, relevance, and risk of bias of studies included in the review. Quality assessment was undertaken independently by two reviewers (RS and SH) and any inconsistencies were resolved by a third reviewer (AP). Due to the small number of included studies, all papers that underwent quality assessment moved to the data collection phase, regardless of their rating. The full quality assessment results can be found in Table A1 in Appendix A. 

### 2.5. Data Collection Process and Summary Measures

Data collection was performed by one reviewer (RS) and checked for accuracy and consistency by a second reviewer (AP). A template was created in Excel for data collection. This included study and participant characteristics (including study type, country, age, gender, sample size), NAFLD diagnostic methodology (e.g., biochemistry, imaging, biopsy), coffee dose and form, and outcome measures. Principal summary measures included odds ratio (OR), *p*-values, or differences in mean coffee intake in relation to severity of NAFLD. Due to the heterogeneity of the included studies, a meta-analysis was not performed. Instead, the effect of coffee consumption on the progression of NAFLD was presented in a narrative summary. Data synthesis was completed to estimate the strength of evidence for the effect, the direction of the effect, and to assess the consistency of the effect across studies. Tables have been used to provide a descriptive summary and explanation of key study characteristics and findings.

## 3. Results

### 3.1. Study Selection

In total, 1782 titles and abstracts were located through the database searches. These were uploaded into Endnote [30] where 902 abstracts were identified as duplicates and removed (Figure 1). The remaining 880 abstracts were then uploaded into Covidence [29], where a further six duplicates were identified and removed. 874 abstracts were screened for eligibility; 44 articles were full text screened, with 39 excluded. Reasons for exclusion included: study involved an intervention not of interest (e.g., focus on green coffee extract); outcome or population group were not of interest (e.g., focus on other liver diseases, i.e., hepatitis C); not a study (e.g., letter to the editor within a journal); trial protocol; and full text could not be accessed by the authors. Five studies only met the inclusion criteria (see Section 2.1) and were included in this review. While this is a small number of eligible studies, this strict exclusion process ensured that all studies were relevant to the specific aim of this review.

### 3.2. Study Characteristics

Key characteristics of the five included studies can be found in Table 1.

#### 3.2.1. Study Design

All studies were cross-sectional in design, however, there were considerable variations found between studies for country, sample size, and age of participants. The studies by Molloy et al. [32] and Bambha et al. [33] were conducted in the USA, the study by Anty et al. [34] in France, the study by Barros et al. [35] in Brazil, and the study by Zelber-Sagi et al. [36] in Israel. The sample size of studies ranged from 112 to 782 participants, with participants’ mean age ranging from 34.7–53.8 years. Whilst all studies included mixed gender populations, females constituted greater than 50% of the population for the studies by Bambha et al. [33], Anty et al. [34], and Barros et al. [35], while in the studies by Molloy et al. [32] and Zelber-Sagi et al. [36] there were an even number of males and females.

#### 3.2.2. Study Population

Study populations varied, with two cohorts sampled from groups of obese patients undergoing bariatric surgery [34,35], one cohort sample from patients attending a specialist hepatology clinic [32], one cohort sampled from the Non-Alcoholic Steatohepatitis Clinic Research Network [33], and the remaining cohort sampled from the Israeli national population registry [36].

#### 3.2.3. Diagnostic Methodology

The study by Zelber-Sagi et al. [36] used ultrasonography as their diagnostic methodology while the remaining four studies used liver biopsies [32,33,34,35]. These liver biopsies were analysed according to the NASH clinical research network (CRN) criteria utilising the Brunt system for grading and staging of steatohepatitis [32,33,34,35].

#### 3.2.4. Measurement of Coffee

To evaluate coffee consumption, the studies by Anty et al. [34], Barros et al. [35], and Zelber-Sagi et al. [36] used face-to-face interviews, the study by Molloy et al. [32] used telephone interviews, and the study by Bambha et al. [33] used a self-administered food frequency questionnaire. Anty et al. [34] and Barros et al. [35] evaluated coffee consumption based on participants’ weekly intake prior to undertaking bariatric surgery, Molloy et al. [32] evaluated coffee consumption based on participants’ typical daily consumption at the time of biopsy, Bambha et al. [33] evaluated coffee consumption based on participants’ usual intake over the previous year, and Zelber-Sagi et al. [36] evaluated coffee consumption based on participants’ usual intake over the previous month. Bambha et al. [33] and Zelber-Sagi et al. [36] reported coffee consumption in number of cups per day, Barros et al. [35] reported coffee consumption in millilitres of coffee consumed per week, while Molloy et al. [32] and Anty et al. [34] reported coffee consumption in terms of the amount of caffeine from coffee ingested, with Anty et al. [34] reporting caffeine in grams per week and Molloy et al. [32] reporting caffeine in milligrams per day.

Molloy et al. [32], Bambha et al. [33], and Barros et al. [35] did not specify the type of coffee assessed, Zelber-Sagi et al. [36] assessed caffeinated and decaffeinated coffee separately but did not report decaffeinated coffee due to low consumption, and Anty et al. [34] assessed espresso and regular filtrated coffee.

### 3.3. Risk of Bias within Studies

The studies by Molloy et al. [32], Bambha et al. [33], Anty et al. [34], and Zelber-Sagi et al. [36] were assigned a positive rating when the Academy of Nutrition and Dietetics Quality Criteria Checklist was completed. However, of these positively rated studies, the studies by Molloy et al. [32] and Bambha et al. [33] did not clearly blind participants or data collectors. The study by Anty et al. [34] did not take into consideration biases and limitations in their conclusion. The study by Barros et al. [35] was assigned a neutral rating, largely influenced by inadequate statistical analysis relevant to the sample size and a conclusion not supported by results, with biases and limitations not taken into consideration. No studies reported the methods of handling losses from the original sample.

### 3.4. Synthesis of Results

Three cross-sectional studies [33,35,36] assessed the relationship between the quantity of coffee intake and degree of hepatic fibrosis, with two finding an inverse relationship between the two factors. Zelber-Sagi et al. found that high coffee consumption, defined as ≥3 cups per day, was associated with a lower proportion of significant fibrosis (*p* = 0.038) [36]. This was supported by a multivariate logistic regression analysis, which associated high coffee consumption with decreased odds for significant fibrosis (OR = 0.49, *p* = 0.041) [36]. Bambha et al. found similar results among NAFLD patients with low insulin resistance, defined as HOMA-IR < 4.3, with any amount of coffee intake reducing the risk of fibrosis (*p* = 0.001) [33]. However, no protective effect of coffee on fibrosis was found among NAFLD patients with high insulin resistance, defined as HOMA-IR ≥ 4.3 (*p* = 0.6) [33]. In comparison with these two studies, Barros et al. found no statistically significant relationship between coffee consumption and NAFLD progression (*p* = 0.812) [35].

Two cross-sectional studies [32,34] assessed the relationship between the quantity of caffeine intake from coffee and degree of hepatic fibrosis. Molloy et al. found a significant difference in coffee caffeine intake between patients with stage 0–1 fibrosis (low fibrosis—Brunt criteria) and patients with stage 2–4 fibrosis (high fibrosis—Brunt criteria) (*p* = 0.016) [32]. A correlation analysis was conducted, which supported a negative relationship between coffee consumption and the severity of hepatic fibrosis (r = −0.215, *p* = 0.035) [32]. Anty et al. reported similar results, however, focused specifically on the amount of caffeine from regular filtrated coffee and espresso coffee [34]. The amount of caffeine from regular filtrated coffee was inversely associated with the severity of fibrosis (*p* = 0.041) [34]. This was supported by a multivariate logistic regression analysis, which identified the consumption of regular filtrated coffee as an independent protective factor for fibrosis (OR = 0.752, *p* = 0.035) [34]. In contrast, a multivariate logistic regression analysis identified a non-significant relationship between the consumption of espresso coffee and fibrosis (OR = 1.009, *p* = 0.916), resulting in the conclusion that regular filtrated coffee, but not espresso, is associated with less severe fibrosis [34].

## 4. Discussion

NAFLD remains the leading cause of chronic liver disease in developed countries [1]. Dietary modification to include a greater consumption of coffee may be a relatively simple intervention for reducing the progression of the disease. To the authors’ knowledge, this is the first systematic review to focus purely on the effect of coffee consumption on the progression of NAFLD. Overall, the evidence presented in this review suggests that higher coffee consumption is inversely associated with the severity of hepatic fibrosis in individuals with NAFLD. However, there were only five studies eligible for inclusion and these were all cross-sectional in design, so results should be interpreted with caution.

The form of coffee and its preparation method may be an important factor in determining the protective effect of coffee on NAFLD progression. Anty et al. found that regular filtrated (filtered) coffee, but not espresso coffee, was associated with a lower level of fibrosis [34]. This is supported by numerous studies that have shown a hepatoprotective role for filtered coffee, while unfiltered coffee may be potentially harmful [16]. Anty et al. proposed that the potentially beneficial compound(s) that exert the hepatoprotective role are present in filtered coffee but not in espresso [34]. Differences in preparation methods certainly result in differences in the composition of coffee [16], with chlorogenic acids being better preserved in filtered coffee compared with espresso coffee, but filtering coffee also removes the beneficial cafestol and kahweol [16]. Europeans tend to favour espresso style coffee whereas in the US, coffee is usually prepared through filtering [34]. In addition, Anty et al. hypothesised that sucrose, may be more commonly added to espresso coffee than filtered coffee, and that this could be deleterious due to the fructose present in the sucrose molecule [34]. Increased severity of hepatic fibrosis in NASH has been linked with fructose consumption [34].

The amount of coffee consumed may also be an important factor in determining any protective effect. All studies examined in this review reported that a higher consumption of coffee was associated with lower severity of hepatic fibrosis [32,33,34,35,36], however, four of the five studies included did not quantify the amount of coffee needed to see an effect [32,33,34,35]. Only the study by Zelber-Sagi et al. quantified the consumption level needed to see an effect, which was determined to be ≥3 cups per day [36]. There is clearly a lack of research surrounding dose of coffee consumption and NAFLD progression to support this, however, a 2019 systematic review looking at coffee consumption and the risk of developing NAFLD determined that an intake of ≥3 cups per day was associated with a lower risk of NAFLD [37]. In comparison, a 2016 systematic review assessing the relationship between coffee consumption and liver cirrhosis determined that an intake of ≥2 cups per day was required to see an effect [15].

In addition to the lack of agreement on a dose of coffee required for hepatoprotection, there is an underlying issue regarding the use of ‘cups per day’ as a form of measurement as cup size can vary significantly and this will have a notable impact on the exact amount of coffee that is consumed. Furthermore, this does not consider any additives such as milk or water that are added to the coffee, subsequently increasing the volume of the beverage, and perhaps interacting with the physiologically active compounds. Inclusion of these additives varies between forms of coffee and, therefore, the focus should be on the amount and concentration of coffee included in the beverage rather than the overall beverage volume. The study by Anty et al., which found that regular filtrated coffee, but not espresso coffee, was associated with lower severity of hepatic fibrosis, compared the consumption of the two forms of coffee in mL per week [34]. However, by doing this they did not take into consideration the addition of additives and, therefore, did not assess the amount of actual coffee being consumed, a major limitation of the study. Further research surrounding the dose of coffee required for a hepatoprotective effect would benefit from utilizing a more reliable, standardised measurement.

There are several mechanistic proposals for why coffee may exert a beneficial effect on NAFLD progression. Coffee is a complex mixture of over one hundred compounds, however, there are considered to be three main compounds that exert physiological effects: caffeine, diterpenes (cafestol and kahweol), and chlorogenic acids [21]. Caffeine may mitigate the progression of hepatic fibrosis by preventing hepatic stellate cell adhesion and activation, stimulating beta-oxidation via an autophagy-lysosomal pathway, and preventing connective tissue growth factor expression by modifying signalling pathways [38]. However, non-caffeine compounds may also play an important role in hepatoprotection [37,38]. Antioxidants chlorogenic acid and uridine diphosphate glucuronosyltransferase may prevent the lipid accumulation in hepatocytes, reduce the inflammatory response, and promote insulin sensitivity [17]. Similarly, diterpenes, such as cafestol and kahweol, can protect the liver via an antioxidant effect by preventing inflammatory reactions through reducing the expression of inflammatory markers [37]. While the exact mechanism of this effect remains unclear, it is clear that the potential hepatoprotective benefit of coffee needs to be further investigated.

The results of this systematic review should be interpreted with caution. In addition to only a small number of studies being eligible for inclusion, the heterogeneity of sample size, population group, and type and quantity of coffee consumption assessed between studies is problematic. All studies included in this review were of a cross-sectional design, providing a low level of evidence, and causality cannot be implied, making results susceptible to residual confounding, over or under estimation of effects, or reverse causality. Further research is needed to prospectively assess the relationship between coffee consumption and NAFLD progression. An additional limitation common across all of the included study methodologies was the use of self-reported dietary data collection techniques, which may contribute to participant bias. It is also important to note that while there may be potential benefits to coffee consumption, there are possible adverse effects of excess coffee intake, including sleep disturbance, anxiety, and irritability [38]. There are certain population groups in which coffee consumption may not be recommended, including pregnant and lactating women as well as individuals with underlying heart and other health conditions [38]. These potential adverse effects and a lack of agreement on the safety threshold for coffee intake must be considered when discussing this topic [38].

Overall, four of the five studies found a statistically significant association between coffee consumption and decreased severity of fibrosis in non-insulin resistant populations. The study by Barros et al. was the exception that found no statistical significance [35]. This study was also the only study to not receive a positive rating during quality assessment [35]. Instead, it was assigned a neutral rating, largely attributed to inadequate statistical analysis relevant to the sample size, a potential factor in its conflicting result [35]. Nevertheless, a conflicting result, particularly in such a small sample of studies, is an underlying limitation of this review. The presence of confounding factors across the studied populations such as age, gender and lifestyle may have an effect on the variation in results found in this review. Furthermore, factors known to influence NAFLD such as insulin resistance and visceral abdominal fat differ across the participants of all five studies and may play a role in the direction of study outcomes.

Despite these limitations, there are a number of strengths in this review. Firstly, this review aimed to address a highly specific research question that was designed to build upon previously conducted research. Secondly, the comprehensive search strategy ensured that the most up to date research evidence was detected. A third strength of this review was that both study selection and quality assessment were undertaken by two authors with any inconsistencies resolved by a third author, greatly increasing the reliability of the results.

Ongoing research in the area of coffee consumption and NAFLD progression would benefit from further investigation into coffee form, preparation method, and quantity needed to see an effect. Experimental and longitudinal studies are warranted to provide evidence for causation, and to reduce confounding variables and various types of bias inherent in observational studies, over a long-term period. Further animal and cell culture studies are also warranted to further explore the biochemical basis for the beneficial effects of coffee in NAFLD patients.

## 5. Conclusions

The results of this review suggest that there is a potentially beneficial effect of coffee consumption on the severity of hepatic fibrosis in NAFLD patients. The optimum quantity and form/preparation method of coffee required to exert this hepatoprotective role remains unclear. Further research in this area is warranted to allow appropriate dietary recommendations to be made.

## Figures and Tables

**Figure 1 nutrients-13-02381-f001:**
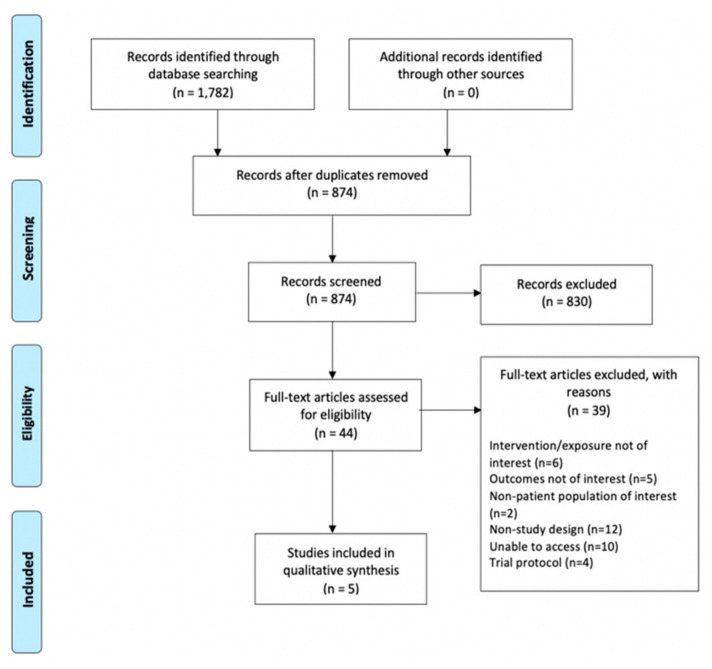
PRISMA flow chart of the study selection process.

**Table 1 nutrients-13-02381-t001:** Summary of characteristics of all studies included in this review.

First Author/Year	Study Details (Study Type, Participants, Country of Study)	Participants (Number, Age Range, Age Mean ± SD, % Female)	Diagnostic Method	Coffee Consumption Measurement	Coffee Form and Dose	Study Results	Final Conclusion	Limitations
Anty, R./ 2012	Cross-sectional study. Morbidly obese patients with NAFLD undergoing bariatric surgery. France.	195 participants. Undefined age range. 39 ± 12.3 years. 82.5%	Liver biopsy †	Face-to-face interview, self-reported consumption in a typical week, pre-surgery	Coffee-containing beverages: espresso; double espresso; filtrated (regular); made with Italian coffee machine; decaffeinated coffee. ** Volume of consumed coffee converted to mg caffeine using purpose-designed converter.	Regular coffee drinkers: caffeine inversely correlated with severity fibrosis (r = −0.26, *p* = 0.041). Consumption of regular coffee an independent protective factor for fibrosis (IR = 0.752, *p* = 0.035). No significant association between espresso coffee consumption and fibrosis (OR = 1.009, *p* = 0.916).	Regular filtrated coffee, but not espresso, was associated with less severe fibrosis	Retrospective. Possible recall/reporting bias.
Bambha, K./2014	Cross-sectional study. Patients with NASH/NAFLD. USA.	782 participants. Undefined age range. 48 ± 12 years. 62% female.	Liver biopsy †	Validated dietary questionnaire (Block Food Frequency Questionnaire), self-reported consumption over the past year	Coffee type not specified; no distinction made between caffeinated and decaffeinated. Coffee consumption reported as average number of cups per day.	Coffee consumers with lower IR (defined as HOMA-IR < 4.3) had lower odds of advanced fibrosis (OR = 0.64, 95% CI, (0.46–0.88), *p* = 0.001). No protective effect of coffee in advanced fibrosis among individuals with higher IR (HOMA-IR ≥ 4.3) (OR = 1.06, 95% CI, {0.87–1.28), *p* = 0.6).	Coffee intake inversely associated with advanced fibrosis in NAFLD patients with lower HOMA-IR. No relationship in NAFLD patients with higher HOMA-IR.	Retrospective. Possible recall/reporting bias. No differentiation between caffeinated and decaffeinated coffee. Truncation of coffee intake at maximum 5 cups per day.
Barros. R./2016	Cross-sectional study. Obese patients with and without NAFLD on biopsy undergoing bariatric surgery. Brazil.	112 participants. Undefined age range. 34.7 ± 7.4 years. 68.6% female.	Liver biopsy †	Face-to-face interview, self-reported consumption in a typical week, pre-surgery	Coffee type not specified, caffeinated only. Coffee reported in mL.	Distribution of NAFLD diagnosis by Tertiles of coffee intake (mean/week): Tertile 1 (0 mL coffee/week) Normal—14.7% Steatosis—23.5% NASH w/o Fibrosis—26.5% NASH w/Fibrosis—35.3% Tertile 2 (860 mL coffee/week) Normal—21.9% Steatosis—15.6% NASH w/o Fibrosis—18.8% NASH w/Fibrosis—43.8% Tertile 3 (3360 mL coffee/week) Normal—24.3% Steatosis—24.3% NASH w/o Fibrosis—16.2% NASH w/Fibrosis—35.1%	Authors reported patients with history of greater coffee consumption exhibited lower frequencies of NASH and fibrosis, although not statistically significant (*p* = 0.812) ~	Retrospective. Possible recall/reporting bias. Lack of blinding.
Molloy, J./2011	Cross-sectional study. NAFLD patients identified from medical records of Brooke Army Medical Center Hepatology Clinic. USA.	306 participants. 18–70 years. 53.8 years (SD not reported). 49% female.	Liver biopsy †	Telephone interview, self-reported consumption at time of biopsy	Coffee type not specified, caffeinated only. Frequency stratified into 9 groups from never to 6 or more cups per day. Caffeine content per unit was calculated to determine daily caffeine intake from coffee.	Mean coffee caffeine intake for NAFLD patients by histological diagnosis: Group 1—negative ultrasound 227.8 mg/day Group 2—bland steatosis/not-NASH 160.3 mg/day Group 3—NASH w/stage 0–1 fibrosis 225.7 mg/day Group 4—NASH w/stage 2–4 fibrosis 152.7 mg/day Significant difference in coffee consumption (*p* = 0.016) between patients with NASH stage 0–1 and NASH stage 2–4. Mean coffee caffeine intake for NASH patients by stage of fibrosis: Stage 1: 255.89 mg/day Stage 2: 170.3 mg/day Stage 4: 122 mg/day Negative relationship between coffee consumption and hepatic fibrosis (r = −0.215, *p* = 0.035)	An increased intake of caffeine from coffee results in a significantly decreased risk of advanced fibrosis.	Retrospective. Possible recall/reporting bias. Lack of blinding of interviewers.
Zelber-Sagi, S./2014	Cross-sectional study. Participants with and without NAFLD (on ultrasound) randomly sampled from First Israeli National Health and Nutrition Survey. Israel.	347 participants. 24–70 years. 50.86 ± 10.35 years. 46.7% female.	Ultrasonography	Face-to-face interview, two questionnaires: (1) specifically assessing regular coffee consumption; (2) detailed semi-quantitative FFQ including coffee consumption	All types caffeinated coffee, decaffeinated not included. Frequency stratified into 8 groups from <once per month to >5 per day.	High coffee consumption (≥3 cups per day) associated with a lower proportion of clinically significant fibrosis ≥ F2 (8.8% vs. 16.3%; *p* = 0.038). Consistently in multivariate logistic regression analysis, high coffee consumption was associated with lower odds for significant fibrosis (OR = 0.49, 95% CI, 0.25–0.97, *p* = 0.041).	Coffee intake inversely associated with liver fibrosis	Retrospective. Possible recall/reporting bias. Absence of liver histology for accurate NAFLD diagnosis.

Notes: † analysed according to NASH clinical research network (CRN) criteria utilizing the Brunt system for grading and staging of steatohepatitis; ** data reported results on espresso and filtrated (regular) coffee only; ~ authors’ conclusion not supported by data.

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
