# Peer review of "Coffee Consumption and the Progression of NAFLD: A Systematic Review"

_nutrients, 2021, doi:10.3390/nu13072381_

Round 1
Reviewer 1 Report
A brief summary (one short paragraph) outlining the aim of the paper and its main contributions.
The aim of the paper is to synthesize the available data on the effect of coffee on the progression of NAFLD. It includes a narrative synthesis based on five cross-sectional studies.
Broad comments highlighting areas of strength and weakness. These comments should be specific enough for authors to be able to respond.
The paper is novel and very well written without grammatical errors. It provides a short and precise overview of the topic, describes the methods in great detail, and provide an adequate discussion of the results.
It succeed to provide a synthesis of the available data on the topic. The systematic approach is a major advantage.
Unfortunately, the main table is unclear and difficult to read.
A weakness of the study is the lack of high quality studies (e.g. RCTs) within the area. With the low level of evidence and the low number of studies, one might question this paper’s relevance.
Specific comments referring to line numbers, tables or figures. Reviewers need not comment on formatting issues that do not obscure the meaning of the paper, as these will be addressed by editors.
Line 137: It could be considered to include the bias assessment in a simple figure, at least as an appendix.
Line 179: Tables are included to give overview or simplify, however, the table does not provide any overview to the reader. The main text actual give a clearer picture. I would suggest that you made the sentences shorter, highlighted the most important information or in other ways improved the table.
Reviewer 2 Report
This systematic review aims to investigate the relationship between coffee consumption and the progression of NAFLD. However, some issues need to be addressed.
- The selection criteria are not clearly described in result 3.1. The author should provide more discrimination of the excluded cases shown in figure 1. The selection process needs more explanation for readers to understand why there were only 5 studies left in the end.
- This paper aims to investigate the relationship between coffee consumption and the progression of NAFLD. However, the included studies showed in table 1, only Bambha, K / 2014, Barros. R / 2016, and Molloy, J / 2011 were indicated as NASH or NAFLD cases.
- The author concluded that four out of the five studies reported a statistically significant relationship between coffee consumption and the severity of fibrosis. However, in result 3.4 it was indicated that Bambha et al found no protective effect of coffee on fibrosis among NAFLD patients with high insulin resistance and Barros et al. found no statistically significant relationship between coffee consumption and NAFLD progression. This makes the conclusion unconvincing.
- Overall, the database is relatively too small which makes it hard to interpret the results.
Reviewer 3 Report
This review addresses a controversial issue with limited experimental support. However, the authors are cautious in their statements and the limitations of the studies are clearly stated. I suggest addressing in the discussion the possible adverse effects of coffee on the nervous system and its contraindication, for example, in people with sleep disorders. A summary table of the limitations of current studies associating coffee consumption and NAFLD would also be helpful.
Round 2
Reviewer 2 Report
The author had addressed all the questions. The manuscript looks good now.